# Padé Activation Units: End-to-end Learning of Flexible Activation Functions in Deep Networks

**Alejandro Molina**[1]**, Patrick Schramowski**[1]**, Kristian Kersting**[1,2]
[1] AI and Machine Learning Group, CS Department, TU Darmstadt, Germany
[2] Centre for Cognitive Science, TU Darmstadt, Germany
`{molina,schramowski,kersting}@cs.tu-darmstadt.de`

## Abstract

The performance of deep network learning strongly depends on the choice of the non-linear activation function associated with each neuron. However, deciding on the best activation is non-trivial, and the choice depends on the architecture, hyper-parameters, and even on the dataset. Typically these activations are fixed by hand before training. Here, we demonstrate how to eliminate the reliance on first picking fixed activation functions by using flexible parametric rational functions instead. The resulting Padé Activation Units (PAUs) can both approximate common activation functions and also learn new ones while providing compact representations. Our empirical evidence shows that end-to-end learning deep networks with PAUs can increase the predictive performance. Moreover, PAUs pave the way to approximations with provable robustness.

`https://github.com/ml-research/pau`

## 1 Introduction

An important building block of deep learning is the non-linearities introduced by the activation functions $f(x)$. They play a major role in the success of training deep neural networks, both in terms of training time and predictive performance. Consider e.g. Rectified Linear Unit (ReLU) due to Nair and Hinton (2010). The demonstrated benefits in training deep networks, see e.g. (Glorot et al., 2011), brought renewed attention to the development of new activation functions. Since then, several ReLU variations with different properties have been introduced such as LeakyReLUs (Maas et al., 2013), ELUs (Clevert et al., 2016), RReLUs (Xu et al., 2015), among others. Another line of research, such as (Ramachandran et al., 2018) automatically searches for activation functions. It identified the Swish unit empirically as a good candidate. However, for a given dataset, there are no guarantees that Swish unit behaves well and the proposed search algorithm is computationally quite demanding.

The activation functions are traditionally fixed and, in turn, impose a set of inductive biases on the network. One attempt to relax this bias are for instance PReLUs (He et al., 2015), where the negative slope is subject to optimization allowing for more flexibility than other ReLU variants. Learnable activation functions generalize this idea. They exploit parameterizations of the activation functions, adapted in an end-to-end fashion to different network architectures and datasets during training. For instance, Maxout (Goodfellow et al., 2013) and Mixout (Zhao et al., 2017) use a fixed set of piecewise linear components and optimized their (hyper-)parameters. Although they are theoretically universal function approximators, they heavily increase the number of parameters of the network and strongly depend on hyper-parameters such as the number of components to realize this potential. Vercellino and Wang (2017) used a meta-learning approach for learning task-specific activation functions (hyperactivations). However, as Vercellino and Wang admit, the implementation of hyperactivations, while easy to express notationally, can be frustrating to implement for generalizability over any given activation network. Recently, Goyal et al. (2019) proposed a learnable activation function based on Taylor approximation and suggested a transformation strategy to avoid exploding gradients. However, relying on polynomials suffers from well-known limitations such as exploding values and a tendency to oscillate (Trefethen, 2012).

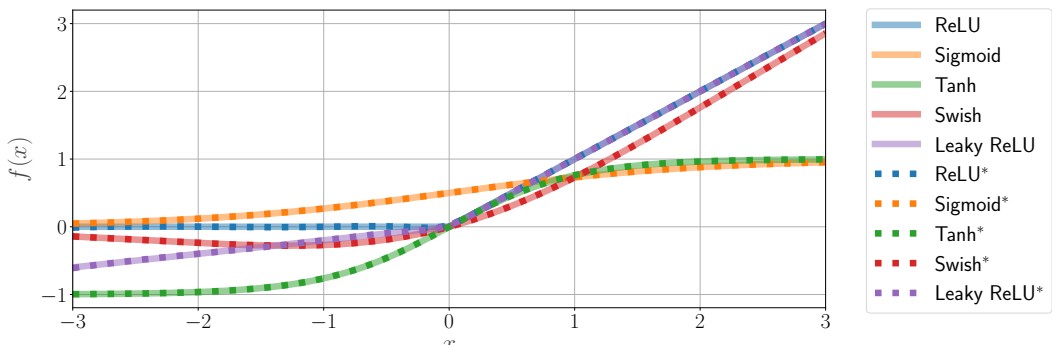

Figure 1: Approximations of common activation functions (ReLU, Sigmoid, Tanh, Swish and Leaky ReLU ($\alpha = 0.20$)) using PAUs (marked with *). As one can see, PAUs can encode common activation functions very well. (Best viewed in color)

As an alternative, we here introduce a learnable activation function based on the Padé approximation, i.e., rational functions. In contrast to approximations for high accuracy hardware implementation of the hyperbolic tangent and the sigmoid activation functions (Hajduk, 2018), we do not assume fixed coefficients. The resulting Padé Activation Units (PAU) can be learned using standard stochastic gradient and, hence, seamlessly integrated into the deep learning stack. PAUs provide more flexibility and increase the predictive performance of deep neural networks, as we demonstrate.

We proceed as follows. We start off by introducing PAUs. Then introduce Padé networks and show that they are universal approximators. Before concluding, we present our empirical evaluation.

## 2 PADÉ ACTIVATION UNITS (PAU)

Our starting point is the set of intuitive assumptions activation functions should ultimately fulfill shown in Tab. 1. The assumptions (i,v) concern the ability of neural networks to approximate functions. Rational functions can fulfill assumptions (i,iv), and our experimental evaluation demonstrates that assumptions (ii,iii,v) also hold.

(i) They must allow the networks to be universal function approximators.

(ii) They should ameliorate gradient vanishing.

(iii) They should be stable.

(iv) They should be parsimonious on the number of parameters.

(v) They should provide networks with high predictive performance.

| Activation | Learnable | Assumptions | | | | |
|---|---|---|---|---|---|---|
| | | i | ii | iii | iv | v |
| **ReLU** | N | Y | Y | Y | | Y |
| **ReLU6** | N | Y | Y | Y | | Y |
| **RReLU** | N | Y | Y | Y | | Y |
| **LReLU** | N | Y | Y | Y | | Y |
| **ELU** | N | Y | Y | Y | | Y |
| **CELU** | N | Y | Y | Y | | Y |
| **Swish** | N | Y | ? | Y | | Y |
| **PReLU** | Y | Y | Y | Y | Y | Y |
| **Maxout** | Y | Y | Y | Y | N | Y |
| **Mixture** | Y | Y | Y | Y | Y | Y |
| **APL** | Y | Y | Y | Y | Y | Y |
| **SReLU** | Y | Y | Y | Y | Y | Y |
| **SLAF** | Y | Y | Y | N | Y | ? |
| **PAU** | Y | Y | Y | Y | Y | Y |

Table 1: (Left) The intuitive assumptions activation functions (AFs) should ultimately fulfill. (Right) Existing ELU, CELU and ReLU like AFS do not fulfill them. Only learnable AFs allow one to tune their shape at training time, ignoring hyper-parameters such as $\alpha$ for LReLU. For Swish, our experimental results do not indicate problems with vanishing gradients. SLAF (Goyal et al., 2019) showed undefined values (iii), and we could not judge their performance (v).

## 2.1 Padé Approximation of Activation Functions

Let us now formally introduce PAUs. Assume for the moment that we start with a fixed activation function $f(x)$. The Padé approximant (Brezinski and Van Iseghem, 1994) is the "best" approximation of $f(x)$ by a rational function of given orders $m$ and $n$. Applied to typical actication functions, Fig. 1 shows that they can be approximated well using rational functions.

More precisely, given $f(x)$, the Padé approximant is the rational function $F(x)$ over polynomials $P(x)$, $Q(x)$ of order $m$, $n$ of the form

$$F(x) = \frac{P(x)}{Q(x)} = \frac{\sum_{j=0}^{m} a_j x^j}{1 + \sum_{k=1}^{n} b_k x^k} = \frac{a_0 + a_1 x + a_2 x^2 + \cdots + a_m x^m}{1 + b_1 x + b_2 x^2 + \cdots + b_n x^n} \ , \tag{1}$$

which agrees with $f(x)$ the best. The Padé approximant often gives a better approximation of a function $f(x)$ than truncating its Taylor series, and it may still work where the Taylor series does not converge. For these reasons, it has been used before in the context of graph convolutional networks (Chen et al., 2018). However, they have not been considered so far for general deep networks. Padé Activation Units (PAUs) go one step further, instead of fixing the coefficients $a_j, b_k$ to approximate a particular activation function, we allow them to be free parameters that can be optimized end-to-end with the rest of the neural network. This allows the optimization process to find the activation function needed at each layer automatically.

The flexibility of Padé is not only a blessing but might also be a curse: it can model processes that contain poles. For a learnable activation function, however, a pole may produce undefined values depending on the input as well as instabilities at learning and inference time. Therefore we consider a restriction, called *safe PAU*, that guarantees that the polynomial $Q(x)$ is not 0, i.e., we avoid poles. In general, restricting $Q(x)$ implies that either $Q(x) \mapsto \mathbb{R}_{>0}$ or $Q(x) \mapsto \mathbb{R}_{<0}$, but as $P(x) \mapsto \mathbb{R}$ we can focus on $Q(x) \mapsto \mathbb{R}_{>0}$ wlog. However, as $\lim_{Q(x) \to 0^+} F(x) \to \infty$ learning and inference become unstable. To fix this, we impose a stronger constraint, namely $Q(x) \geq q \gg 0$. In this work, $q = 1$, i.e., $\forall x : Q(x) \geq 1$, preventing poles and allowing for safe computation on $\mathbb{R}$:

$$F(x) = \frac{P(x)}{Q(x)} = \frac{\sum_{j=0}^{m} a_j x^j}{1 + |\sum_{k=1}^{n} b_k x^k|} = \frac{a_0 + a_1 x + a_2 x^2 + \cdots + a_m x^m}{1 + |b_1 x + b_2 x^2 + \cdots + b_n x^n|} \ . \tag{2}$$

Other values for $q \in (0,1)$ might still be interesting, as they could provide gradient amplification due to the partial derivatives having $Q(X)$ in the denominator. However, we leave this for future work.

## 2.2 Learning Safe Padé Approximations using Backpropagation

In contrast to the standard way of fitting Padé approximants where the coefficients are found via derivatives and algebraic manipulation against a given function, we optimize their polynomials via backpropagation and (stochastic) gradient descent. To do this, we have to compute the gradients with respect to the parameters $\frac{\partial F}{\partial a_j}$, $\frac{\partial F}{\partial b_k}$ as well as the gradient for the input $\frac{\partial F}{\partial x}$. A simple alternative is to implement the forward pass as described in Eq. 2, and let automatic differentiation do the job. To be more efficient, however, we can also implement PAUs directly in CUDA (Nickolls et al. (2008)), and for this we need to compute the gradients ourselves:

$$\frac{\partial F}{\partial x} = \frac{\partial P(x)}{\partial x} \frac{1}{Q(x)} - \frac{\partial Q(x)}{\partial x} \frac{P(x)}{Q(x)^2}, \quad \frac{\partial F}{\partial a_j} = \frac{x^j}{Q(x)} \quad \text{and} \quad \frac{\partial F}{\partial b_k} = -x^k \frac{A(X)}{|A(X)|} \frac{P(X)}{Q(x)^2} \ ,$$

where $\frac{\partial P(x)}{\partial x} = a_1 + 2a_2 x + \cdots + m a_m x^{m-1}$ , $\frac{\partial Q(x)}{\partial x} = \frac{A(X)}{|A(X)|} \left( b_1 + 2b_2 x + \cdots + n b_n x^{n-1} \right)$ , $A(X) = b_1 x + b_2 x^2 + \cdots + b_n x^n$ , and $Q(x) = 1 + |A(X)|$. Here we reuse the expressions to reduce computations. To avoid divisions by zero when computing the gradients, we define $\frac{z}{|z|}$ as the sign of $z$. With the gradients at hand, PAUs can seamlessly be placed together with other modules onto the differentiable programming stack.

## 3 Padé Networks

Having PAUs at hand, one can define Padé networks as follows: Padé networks are feedforward networks with PAU activation functions that may include convolutional and residual architectures

with pooling layers. To use Padé networks effectively, one simply replaces the standard activation functions in a neural network by PAUs and then proceed to optimize all the parameters and use the network as usual. However, even if every PAU contains a low number of parameters (coefficients $a_j, b_k$), in the extreme case, learning one PAU per neuron may considerably increase the complexity of the networks and in turn the learning time. To ameliorate this and inspired by the idea of weight-sharing as introduced by Teh and Hinton (2001), we propose to learn one PAU per layer. Therefore we only add $\phi$ many parameters, where $\phi = L * (m + n)$ and $L$ is the number of activation layers in the network. In our experiments we set $\phi = 10L$, a rather small number of parameters (iv).

The last step missing before we can start the optimization process is to initialize the coefficients of the PAUs. Surely, one can do random initialization of the coefficients and allow the optimizer to train the network end-to-end. However, we obtained better results after initializing all PAUs with coefficients that approximate standard activation functions. For a discussion on how to obtain different PAU coefficients, we refer to Sec. A.1.

Before evaluating Padé Networks empirically, let us touch upon their expressivity and how to sparsify them.

## 3.1 PADÉ NETWORKS ARE UNIVERSAL FUNCTION APPROXIMATORS

A standard multi-layer perceptron (MLP) with enough hidden units and non-polynomial activation functions is a universal approximator, see e.g. (Hornik et al., 1989; Leshno et al., 1993). Padé Networks are also universal approximators.

**Theorem 1.** *Let $\rho \colon \mathbb{R} \to \mathbb{R}$ be a PAU activation function. Let $\mathcal{N}^\rho$ represent the class of neural networks with activation function $\rho$. Let $K \subseteq \mathbb{R}^n$ be compact. Then $\mathcal{N}^\rho$ is dense in $C(K)$.*

The proof holds for both PAUs and safe PAUs as it makes no assumptions on the form of the denominator $Q(x)$, and is a direct application of the following propositions:

**Proposition 1.** *(From Theorem 1.1 in (Kidger and Lyons, 2019)) Let $\rho \colon \mathbb{R} \to \mathbb{R}$ be any continuous function. Let $\mathcal{N}_n^\rho$ represent the class of neural networks with activation function $\rho$, with $n$ neurons in the input layer, one neuron in the output layer, and one hidden layer with an arbitrary number of neurons. Let $K \subseteq \mathbb{R}^n$ be compact. Then $\mathcal{N}_n^\rho$ is dense in $C(K)$ if and only if $\rho$ is non-polynomial.*

**Proposition 2.** *(From Theorem 3.2 in (Kidger and Lyons, 2019)) Let $\rho \colon \mathbb{R} \to \mathbb{R}$ be any continuous function which is continuously differentiable at at least one point, with nonzero derivative at that point. Let $K \subseteq \mathbb{R}^n$ be compact. Then $\mathcal{NN}_{n,m,n+m+2}^\rho$ is dense in $C(K; \mathbb{R}^m)$.*

*Proof.* Let $\rho(x) = P(x)/Q(x)$, we have to consider the following two cases:

Case 1: $Q(x) \neq 1$, by definition, $\rho(x)$ is non-polynomial. Then by proposition 1, we get that $\mathcal{N}_n^\rho$ is dense in $C(K)$.

Case 2: $Q(x) = 1$, here $\rho(x) = \sum_{j=0}^m a_j x^j$, is polynomial, continuous and continuously differentiable in $\mathbb{R}$. Let any $a_{j>0} > 0$, then there exists a point $\alpha \in \mathbb{R}$ such that $\rho'(\alpha) \neq 0$ and then by proposition 2, we get that $\mathcal{NN}_{n,m,n+m+2}^\rho$ is dense in $C(K; \mathbb{R}^m)$.

$\square$

## 3.2 SPARSE PADÉ NETWORKS AND RANDOMIZED PAUS

Padé Networks can $\epsilon$-approximate neural networks with ReLU activations (Telgarsky, 2017). This implies that by using PAUs, we are embedding a virtual network into the networks we want to use. This, in turn, is the operating assumption of the lottery ticket hypothesis due to Frankle and Carbin (2019). Thus, we expect that lottery ticket pruning can find well-performing Padé networks that are smaller than their original counterparts while reducing inference time and potentially improving the predictive performance.

Generally, overfitting is an important practical concern when training neural networks. Usually, we can apply regularization techniques such as Dropout (Srivastava et al., 2014). Unfortunately, although each PAU approximates a small ReLU network section, we do not have access to the internal representation of this virtual network. Therefore, we can not regularize the activation function via

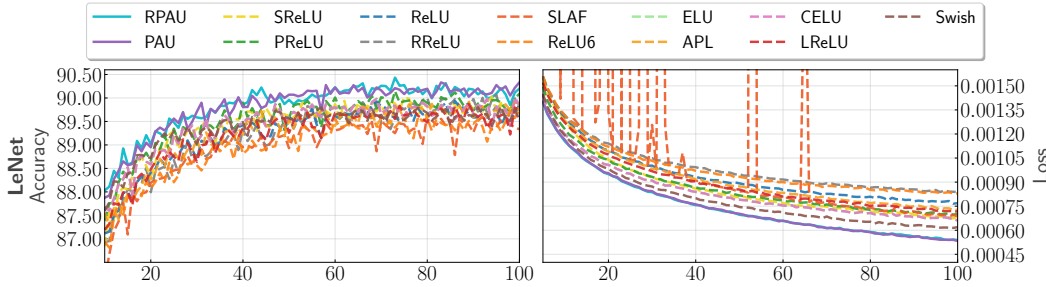

Figure 2: PAU compared to baseline activation function units over 5 runs on Fashion-MNIST using the LeNet architecture: (left) mean test-accuracy (the higher, the better) and (right) mean train-loss (the lower, the better). As one can see, PAU outperforms all baseline activations and enable the networks to achieve a lower loss during training compared to all baselines. (Best viewed in color)

standard Dropout. An alternative for regularizing activation functions was introduced in Randomized Leaky ReLUs (RReLUs, Xu et al. (2015)), where the negative slope parameter is sampled uniformly on a range. This makes the activation function behave differently at training time for every input $x$, forwarding and backpropagating according to $x$ and the sampled noise.

We can employ a similar technique to make PAUs resistant to overfitting. Consider a PAU with coefficients $\mathbf{C} = \{a_0, \cdots, a_m, b_0, \cdots, b_n\}$. We can introduce additive noise during training into each coefficient $c_i \in \mathbf{C}$ for every input $x_j$ via $c_{i,j} = c_i + z_{i,j}$ where $z_{i,j} \sim U(l_i, u_i)$, $l_i = (1 - \alpha\%) * c_i$ and reciprocally $u_i = (1 + \alpha\%) * c_i$. This results in Randomized PAU (RPAU):

$$R(x_j) = \frac{c_{0,j} + c_{1,j}x + c_{2,j}x^2 + \cdots + c_{m,}x^m}{1 + |c_{m+1}x + c_{m+2}x^2 + \cdots + c_{m+n}x^n|} . \tag{3}$$

We compute the gradients as before and simply replace the coefficients by their noisy counterparts.

## 4 EXPERIMENTAL EVALUATION

Our intention here is to investigate the behavior and performance of PAUs as well as to compare them to other activation functions using standard deep neural networks. All our experiments are implemented in PyTorch with PAU implemented in CUDA, and were executed on an NVIDIA DGX-2 system. In all experiments, we initialized PAUs with coefficients that approximate LeakyReLUs for a rational function of order $m = 5, n = 4$. In all experiments except for ImageNet, we report the mean of 5 runs initialized with different seeds for the accuracy on the test-set after training. And, we compared **PAU** to the following activation functions: **ReLU**, **ReLU6**, **Leaky ReLU** (LReLU), **Random ReLU** (RReLU), **ELU**, **CELU**, **Swish**, **Parametric ReLU** (PReLU), **Maxout**, **Mixture of activations** (Mixture), **SLAF**, **APL** and **SReLU**. For details on all the activation functions, we refer to Appendix A.2. As datasets we considered **MNIST**, **Fahion-MNIST**, **CIFAR-10** and **ImageNet**.

### 4.1 EMPIRICAL RESULTS ON MNIST AND FASHION-MNIST BENCHMARKS

First we evaluated PAUs on MNIST (LeCun et al., 2010) and Fashion-MNIST (Xiao et al., 2017) using two different architectures: LeNet (LeCun et al., 1998) and VGG-8 (Simonyan and Zisserman, 2015). For more details on the architectures, learning settings, and results, we refer to Sec. A.3.

As can be seen in Fig. 2 and Tab. 2, PAU outperformed on average the baseline activation functions on every network in terms of predictive performance. Moreover, the results are stable on different runs (c.f. mean $\pm$ std). PAUs also enable the networks to achieve a lower loss during training compared to all baselines on all networks. Actually, PAU achieved the best results on both datasets and on Fashion-MNIST it provides the best results for both architectures. As expected, reducing the bias is beneficial in this experiment. Comparing the baseline activation functions on the MNIST dataset and the different architectures, there is no clear choice of activation that achieves the best performance. However, PAU always matches or even outperforms the best performing baseline activation function. This shows that a learnable activation function relieves the network designer of having to commit to a

Table 2: Performance comparison of activation functions on MNIST and Fashion-MNIST (the higher, the better) on two common deep architectures. Shown are the results averaged over 5 reruns as well as the top result among these 5 runs. The best ("●") and runner-up ("○") results per architecture are **bold**. As one can see, PAUs consistently outperform the other activation functions on average and yields the top performance on each dataset.

| | VGG-8 mean ± std | best | LeNet mean ± std | best | VGG-8 mean ± std | best | LeNet mean ± std | best |
|---|---|---|---|---|---|---|---|---|
| | **MNIST** | | | | **Fashion-MNIST** | | | |
| **ReLU** | $99.17 \pm 0.10$ | 99.30 | $99.17 \pm 0.05$ | 99.25 | $89.11 \pm 0.43$ | 89.69 | $89.86 \pm 0.32$ | 90.48 |
| **ReLU6** | $99.28 \pm 0.04$ | 99.31 | $99.09 \pm 0.09$ | 99.22 | $89.87 \pm 0.62$ | 90.38 | $89.74 \pm 0.27$ | 89.96 |
| **LReLU** | $99.13 \pm 0.11$ | 99.27 | $99.10 \pm 0.06$ | 99.22 | $89.37 \pm 0.30$ | 89.74 | $89.74 \pm 0.24$ | 90.02 |
| **RReLU** | $99.16 \pm 0.13$ | 99.28 | $99.20 \pm 0.13$ | ○**99.38** | $88.46 \pm 0.85$ | 89.32 | $89.74 \pm 0.19$ | 89.88 |
| **ELU** | $99.15 \pm 0.09$ | 99.28 | $99.15 \pm 0.06$ | 99.22 | $89.65 \pm 0.33$ | 90.06 | $89.84 \pm 0.47$ | 90.25 |
| **CELU** | $99.15 \pm 0.09$ | 99.28 | $99.15 \pm 0.06$ | 99.22 | $89.65 \pm 0.33$ | 90.06 | $89.84 \pm 0.47$ | 90.25 |
| **Swish** | $99.10 \pm 0.06$ | 99.20 | $99.19 \pm 0.09$ | 99.29 | $88.54 \pm 0.59$ | 89.36 | $89.54 \pm 0.22$ | 89.89 |
| **PReLU** | $99.16 \pm 0.09$ | 99.25 | $99.14 \pm 0.09$ | 99.24 | $88.82 \pm 0.51$ | 89.54 | $90.09 \pm 0.22$ | ○**90.29** |
| **SLAF** | — | — | — | — | $90.60 \pm 0.00$ | 90.60 | $89.33 \pm 0.28$ | 89.80 |
| **APL** | ○**99.35 ± 0.11** | ●**99.50** | $99.18 \pm 0.10$ | 99.33 | ●**91.41 ± 0.48** | ●**92.25** | $89.72 \pm 0.30$ | 90.01 |
| **SReLU** | $99.15 \pm 0.03$ | 99.20 | $99.13 \pm 0.14$ | 99.27 | $89.65 \pm 0.42$ | 90.31 | $89.83 \pm 0.30$ | 90.28 |
| **PAU** | $99.30 \pm 0.05$ | ○**99.40** | ○**99.21 ± 0.04** | 99.26 | ○**91.25 ± 0.18** | ○**91.56** | ●**90.33 ± 0.15** | ●**90.62** |
| **RPAU** | ●**99.35 ± 0.04** | 99.38 | ●**99.26 ± 0.11** | ●**99.42** | $91.23 \pm 0.15$ | 91.41 | ○**90.20 ± 0.11** | ○**90.29** |

potentially suboptimal choice. Moreover, Fig. 2 also shows that PAU is more stable than SLAF. This is not unexpected as Taylor approximations tend to oscillate and overshoot (Trefethen, 2012). We also observed undefined values at training time for SLAF; therefore, we do not compare against it in the following experiments. Finally, when considering the number of parameters used by PAU, we can see that they are very efficient. The VGG-8 network uses 9.2 million parameters, PAU here uses 50 parameters, and for LeNet, the network uses 0.5 million parameters while PAU uses only 40.

In summary, this shows that PAUs are stable, parsimonious and can improve the predictive performance of deep neural networks (iii,iv,v).

## 4.2 LEARNED ACTIVATION FUNCTIONS ON MNIST AND FASHION-MNIST

When looking at the activation functions learned from the data, we can see that the PAU family is flexible yet presents similarities to standard functions. In particular, Fig. 3 illustrates that some of the learned activations seem to be smoothed versions of Leaky ReLUs, since V-shaped activations are simply Leaky ReLUs with negative $\alpha$ values. This is not surprising as we initialize PAUs with coefficients that match Leaky ReLUs. Finding different initialization and optimization parameters is left as future work. In contrast, when learning piecewise approximations of the same activations using Maxout, we would require a high $k$. This significantly increases the number of parameters of the network. This again provides more evidence in favor of PAUs being flexible and parsimonious (iv). SLAF produced undefined values during training on all networks except Fashion-MNIST where LeNet finished 4 runs and VGG only one run.

## 4.3 EMPIRICAL RESULTS ON CIFAR-10

After investigating PAU on MNIST and Fashion-MNIST, we considered a more challenging setting: CIFAR-10 (Krizhevsky et al. (2009)). We also considered other *learnable* activation functions, namely Maxout (k=2) and Mixture of activations (Id and ReLU) as well as another popular deep network architectures: MobileNetV2 (Sandler et al., 2018), ResNet101 (He et al., 2016) and DenseNet121 (Huang et al., 2017). For more details on the learning settings and results, we refer to Sec. A.4.

Let us start by considering the results for VGG-8 and MobileNetV2 on CIFAR-10. These networks are the smallest of this round of experiments and, therefore, could benefit more from bias reduction. Indeed, we can see in Tab. 3 that both networks take advantage of learnable activation functions, i.e., Maxout, PAU, and RPAU. As expected, adding more capacity to VGG-8 helps and this is what Maxout is doing. Moreover, even if Mixtures do not seem to provide a significant benefit on VGG-8,

Table 3: Performance comparison of activation functions on CIFAR-10 (the higher, the better) on four state-of-the-art deep neural architectures. Shown are the results averaged over 5 reruns as well as the top result among these 5 runs. The best ("●") and runner-up ("○") results per architecture are **bold**. As one can see, PAUs are either in the lead or close to the best. ("***") are experiments that did not finish on time.

| | VGG-8 mean ± std | best | MobileNetV2 mean ± std | best | ResNet101 mean ± std | best | DenseNet121 mean ± std | best |
|---|---|---|---|---|---|---|---|---|
| **ReLU** | $92.32 \pm 0.16$ | 92.58 | $91.51 \pm 0.28$ | 91.82 | $95.07 \pm 0.17$ | 95.36 | ○**95.36 ± 0.18** | ○**95.63** |
| **ReLU6** | $92.36 \pm 0.06$ | 92.47 | $91.30 \pm 0.23$ | 91.57 | $95.11 \pm 0.24$ | 95.29 | $95.33 \pm 0.14$ | 95.46 |
| **LReLU** | $92.43 \pm 0.14$ | 92.65 | $91.94 \pm 0.12$ | 92.08 | $95.08 \pm 0.19$ | 95.29 | ●**95.42 ± 0.17** | ●**95.65** |
| **RReLU** | $92.32 \pm 0.07$ | 92.42 | ○**94.66 ± 0.16** | ○**94.94** | $95.21 \pm 0.23$ | ○**95.51** | $95.00 \pm 0.12$ | 95.14 |
| **ELU** | $91.24 \pm 0.09$ | 91.33 | $90.43 \pm 0.14$ | 90.61 | $94.04 \pm 0.14$ | 94.24 | $90.78 \pm 0.29$ | 91.23 |
| **CELU** | $91.24 \pm 0.09$ | 91.33 | $90.69 \pm 0.27$ | 90.97 | $93.80 \pm 0.36$ | 94.25 | $90.88 \pm 0.19$ | 91.08 |
| **PReLU** | $92.22 \pm 0.26$ | 92.51 | $93.54 \pm 0.45$ | 93.95 | $94.15 \pm 0.39$ | 94.50 | $94.98 \pm 0.16$ | 95.15 |
| **Swish** | $91.58 \pm 0.18$ | 91.86 | $92.04 \pm 0.13$ | 92.21 | $91.83 \pm 1.61$ | 92.84 | $93.04 \pm 0.16$ | 93.32 |
| **Maxout** | ●**93.03 ± 0.11** | ●**93.23** | $94.41 \pm 0.10$ | 94.54 | $95.11 \pm 0.13$ | 95.23 | *** | *** |
| **Mixture** | $91.86 \pm 0.14$ | 92.06 | $94.06 \pm 0.16$ | 94.25 | $94.50 \pm 0.25$ | 94.71 | $93.33 \pm 0.17$ | 93.59 |
| **APL** | $91.63 \pm 0.13$ | 91.82 | $93.62 \pm 0.64$ | 94.50 | $94.12 \pm 0.36$ | 94.50 | $94.45 \pm 0.23$ | 94.78 |
| **SReLU** | ○**92.66 ± 0.27** | ○**93.13** | $94.03 \pm 0.11$ | 94.25 | ○**95.24 ± 0.13** | 95.38 | $94.77 \pm 0.24$ | 95.20 |
| **PAU** | $92.51 \pm 0.16$ | 92.70 | $94.57 \pm 0.21$ | 94.90 | $95.16 \pm 0.13$ | 95.28 | $95.03 \pm 0.07$ | 95.16 |
| **RPAU** | $92.50 \pm 0.09$ | 92.62 | ●**94.82 ± 0.21** | ●**95.13** | ●**95.34 ± 0.13** | ●**95.54** | $95.27 \pm 0.10$ | 95.41 |

they do help in MobileNetV2. Here, we see again that PAU and RPAU are either in the lead or close to the best when it comes to predictive performance, without having to make a choice apriori.

Now, let us have a look at the performance of PAU and RPAU on the larger networks ResNet101 and DenseNet121. As these networks are so expressive, we do not expect the flexibility of the learnable activation functions to have a big impact on the performance. Tab. 3 confirms this. Nevertheless, they are still competitive and their performance is stable as shown by the standard deviation. On ResNet101, PAUs actually provided the top performance.

## 4.4 FINDING SPARSE PADÉ NETWORKS

As discussed in Sec. 3.2, using PAUs in a network is equivalent to introducing virtual networks of ReLUs in the middle of the network, effectively adding virtual depth to the networks. Therefore, we also investigated whether pruning can help one to unmask smaller sub-networks whose performance is similar to the original network. In a sense, we are removing blocks of the real network as they get replaced by the virtual network. Here, we only do pruning on the convolutional layers. For details about the algorithm and hyper-parameters, wee refer to Sec. A.4.3.

Specifically, we compared PAU against the best activation functions for the different architectures. However, we discarded Maxout, as instead of pruning it introduces many more parameters into

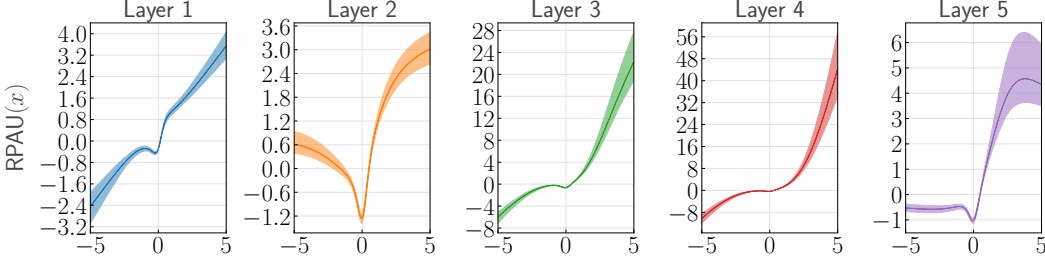

Figure 3: Estimated activation functions after training the VGG-8 network with RPAU on Fashion-MNIST. The center line indicates the PAU while the surrounding area indicates the space of the additive noise in RPAUs. As one can see, the PAU family differs from common activation functions but capture characteristics of them. (Best viewed in color)

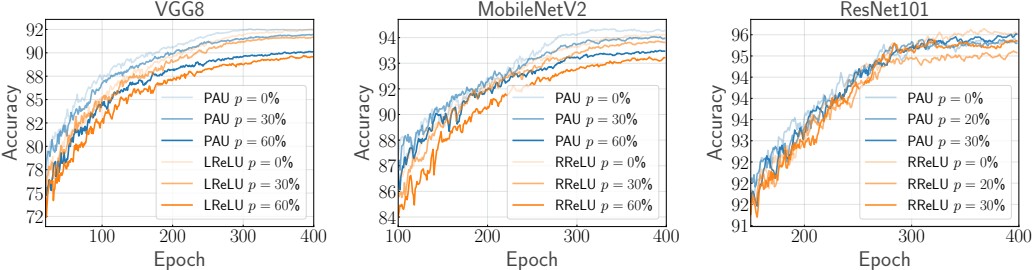

Figure 4: Comparison of the predictive accuracy (higher is better) for the architectures VGG-8, MobileNetV2 and ResNet101 between PAU and the best activation functions according to Tab. 3. PAU is consistently better. On ResNet101 PAU is not affected by the increase pruning pressure. Furthermore, PAU enables the ResNet101 subnetwork, pruned by 30%, to achieve a higher accuracy compared to all pruned and not pruned networks. (Best viewed in color)

the network defeating the original purpose. As one can see in Fig. 4, pruning on the already size-optimized networks VGG-8 and MobileNetV2 has an effect on the predictive performance. However, the performance of PAU remains above the other activation functions despite the increase in pruning pressure. In contrast, when we look at ResNet101, we see that the performance of PAU is not influenced by pruning, showing that indeed we can find sparse Padé network without major loss in accuracy. And what is more, PAU enables the ResNet101 subnetwork, pruned by 30%, to achieve a higher accuracy compared to all pruned and not pruned networks.

## 4.5 EMPIRICAL RESULTS ON IMAGENET

Finally, we investigated the performance on a much larger dataset, namely ImageNet (Russakovsky et al. (2015)) used to train MobileNetV2. As can be seen in Fig. 5 and Tab. 4, PAU and Swish clearly dominate in performance (v). PAU leads in top-1 accuracy and Swish in top-5 accuracy. Moreover, both PAU and Swish show faster learning compared to the other activation functions.

Furthermore, we argue that the rapid learning rate of PAU in all the experiments indicate that they do not exhibit vanishing gradient issues (ii).

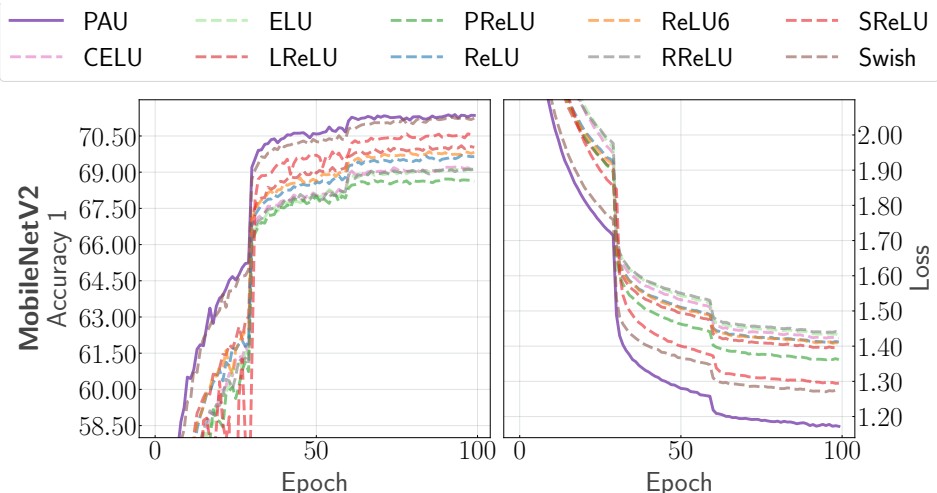

Figure 5: MobileNetV2 top-1 test accuracy on the left (higher is better) and training loss on the right (lower is better) for multiple activation functions in ImageNet. PAU achieves higher accuracy and lower loss values in fewer epochs. (Best viewed in color)

| MobileNetV2 | ReLU | ReLU6 | LReLU | RReLU | ELU | CELU | PReLU | Swish | SReLU | **PAU** |
|---|---|---|---|---|---|---|---|---|---|---|
| **Acc@1** | 69.65 | 69.83 | 70.03 | 69.12 | 69.13 | 69.17 | 68.61 | ∘**71.24** | 70.62 | •**71.35** |
| **Acc@5** | 89.09 | 89.34 | 89.26 | 88.80 | 88.46 | 88.59 | 88.51 | •**89.95** | 89.59 | ∘**89.85** |

Table 4: MobileNetV2 top-1 and top-5 accuracies in ImageNet (higher is better) for different activations. Best ("•") and runner-up ("∘") are shown in **bold**. PAU is the best in top-1 accuracy and runner-up for top-5.

## 4.6 SUMMARIZED RESULTS

Finally, we compare the PAU family of activation functions to all the other activation functions and we aggregate the number of occurrences where PAU performed better or worse in comparison. The aggregate results can be found in Tab. 5. These results are the aggregates of all the experiments on all datasets and all architectures. As we can see, the PAU family is very competitive.

| Baselines | ReLU | ReLU6 | LReLU | RReLU | ELU | CELU | PReLU | Swish | Maxout | Mixture | APL | SReLU |
|---|---|---|---|---|---|---|---|---|---|---|---|---|
| PAU/RPAU >= Baseline | 34 | 35 | 34 | 33 | 40 | 40 | 39 | 41 | 9 | 20 | 32 | 33 |
| PAU/RPAU < Baseline | 8 | 7 | 8 | 9 | 2 | 2 | 3 | 1 | 6 | 0 | 7 | 8 |

Table 5: The number of models on which PAU and RPAU outperforms or underperforms each baseline activation function we compared against in our experiments.

To summarize, PAUs satisfy all the assumptions (i-v). They allow the network to be universal function approximators (i) as shown by theorem 1. They present a fast and stable learning behavior (ii, iii) as shown in Figs. (5,6,7,8). The number of parameters introduced by PAUs is minimal in comparison to the size of the networks. In our experiments, we add 10 parameters per layer, showing that PAUs are parsimonious (iv). Finally, they allow deep neural networks to provide high predictive performance (v) as shown in Tab. 5.

## 5 CONCLUSIONS

We have presented a novel learnable activation function, called Padé Activation Unit (PAU). PAUs encode activation functions as rational functions, trainable in an end-to-end fashion using backpropagation. This makes it easy for practitioners to replace standard activation functions with PAU units in any neural network. The results of our empirical evaluation for image classification demonstrate that PAUs can indeed learn new activation functions and in turn novel neural networks that are competitive to state-of-the-art networks with fixed and learned activation functions. Actually, across all activation functions and architectures, Padé networks are among the top performing networks. This clearly shows that the reliance on first picking fixed, hand-engineered activation functions can be eliminated and that learning activation functions is actually beneficial and simple. Moreover, our results provide the first empirically evidence that the open question "Can rational functions be used to design algorithms for training neural networks?" raised by Telgarsky (2017) can be answered affirmatively for common deep architectures.

Our work provides several interesting avenues for future work. One should explore more the space between safe and unsafe PAUs, in order to gain even more predictive power. Most interestingly, since Padé networks can be reduced to ReLU networks. one should explore globally optimal training (Arora et al., 2018) as well as provable robustness (Croce et al., 2019) of Padé approximations of general deep networks.

**Acknowledgments.** PS and KK were supported by funds of the German Federal Ministry of Food and Agriculture (BMEL) based on a decision of the Parliament of the Federal Republic of Germany via the Federal Office for Agriculture and Food (BLE) under the innovation support program, project "DePhenS" (FKZ 2818204715).

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

# A APPENDIX

## A.1 INITIALIZATION COEFFICIENTS

As show in Table 6 we compute initial coefficients for PAU approximations to different known activation functions. We predefined the orders to be [5,4] and for Sigmoid, Tanh and Swish, we have computed the Padé approximant using the standard techniques. For the different variants of PRelu, LeakyRelu and Relu we optimized the coefficients using least squares over the line range between [-3,3] in steps of 0.000001.

| | Sigmoid | Tanh | Swish | ReLU | LReLU(0.01) | LReLU(0.20) | LReLU(0.25) | LReLU(0.30) | LReLU(-0.5) |
|---|---|---|---|---|---|---|---|---|---|
| $a_0$ | $1/2$ | 0 | 0 | 0.02996348 | 0.02979246 | 0.02557776 | 0.02423485 | 0.02282366 | 0.02650441 |
| $a_1$ | $1/4$ | 1 | $1/2$ | 0.61690165 | 0.61837738 | 0.66182815 | 0.67709718 | 0.69358438 | 0.80772912 |
| $a_2$ | $1/18$ | 0 | $b/4$ | 2.37539147 | 2.32335207 | 1.58182975 | 1.43858363 | 1.30847432 | 13.56611639 |
| $a_3$ | $1/144$ | $1/9$ | $3b^2/56$ | 3.06608078 | 3.05202660 | 2.94478759 | 2.95497990 | 2.97681599 | 7.00217900 |
| $a_4$ | $1/2016$ | 0 | $b^3/168$ | 1.52474449 | 1.48548002 | 0.95287794 | 0.85679722 | 0.77165297 | 11.61477781 |
| $a_5$ | $1/60480$ | $1/945$ | $b^4/3360$ | 0.25281987 | 0.25103717 | 0.23319681 | 0.23229612 | 0.23252265 | 0.68720375 |
| $b_1$ | 0 | 0 | 0 | 1.19160814 | 1.14201226 | 0.50962605 | 0.41014746 | 0.32849543 | 13.70648993 |
| $b_2$ | $1/9$ | $4/9$ | $3b^2/28$ | 4.40811795 | 4.39322834 | 4.18376890 | 4.14691964 | 4.11557902 | 6.07781733 |
| $b_3$ | 0 | 0 | 0 | 0.91111034 | 0.87154450 | 0.37832090 | 0.30292546 | 0.24155603 | 12.32535229 |
| $b_4$ | $1/10008$ | $1/63$ | $b^4/1680$ | 0.34885983 | 0.34720652 | 0.32407314 | 0.32002850 | 0.31659365 | 0.54006880 |

Table 6: Initial coefficients to approximate different activation functions.

## A.2 LIST OF ACTIVATION FUNCTIONS

For our experiments, we compare against the following activation functions with their respective parameters.

- **ReLU** (Nair and Hinton, 2010): $y = \max(x, 0)$
- **ReLU6** (Krizhevsky and Hinton, 2010): $y = \min(\max(x, 0), 6)$, a variation of ReLU with an upper bound.
- **Leaky ReLU** (Maas et al., 2013): $y = \max(0, x) + \alpha * \min(0, x)$ with the negative slope, which is defined by the parameter $\alpha$. Leaky ReLU enables a small amount of information to flow when $x < 0$.
- **Random ReLU** (Xu et al., 2015): a randomized variation of Leaky ReLU.
- **ELU** (Clevert et al., 2016): $y = \max(0, x) + \min(0, \alpha * (\exp(x) - 1))$.
- **CELU** (Barron, 2017): $y = \max(0, x) + \min(0, \alpha * (\exp(x/\alpha) - 1))$.
- **Swish** (Ramachandran et al., 2018): $y = x * \text{sigmoid}(x)$ , which tends to work better than ReLU on deeper models across a number of challenging datasets.
- **Parametric ReLU** (PReLU) (He et al., 2015) $y = \max(0, x) + \alpha * \min(0, x)$ , where the leaky parameter $\alpha$ is a learn-able parameter of the network.
- **Maxout** (Goodfellow et al., 2013): $y = \max(z_{ij})$, where $z_{ij} = x^T W_{...ij} + b_{ij}$ , and $W \in R^{d \times m \times k}$ and $b \in R^{m \times k}$ are learned parameters.
- **Mixture of activations** (Manessi and Rozza, 2018): a combination of weighted activation functions *e.g.* {id, ReLU}, where the weight is a learnable parameter of the network.
- **SLAF** (Goyal et al., 2019): a learnable activation function based on a Taylor approximation.
- **APL** (Agostinelli et al., 2015): a learnable piecewise linear activation function.
- **SReLU** (Jin et al., 2016): a learnable S-shaped rectified linear activation function.

## A.3 DETAILS OF THE MNIST AND FASHION-MNIST EXPERIMENT

### A.3.1 NETWORK ARCHITECTURES

Here we describe the architectures for the networks VGG and LeNet, along with the number of trainable parameters. The number of parameters of the activation function is reported for using PAU. Common not trainable activation functions do not have trainable parameters. PReLU has one

trainable parameter. In total the VGG network as 9224508 parameters with 50 for PAU, and the LeNet network has 61746 parameters with 40 for PAU.

| No. | VGG | # params | LeNet | # params |
|---|---|---|---|---|
| 1 | Convolutional 3x3x64 | 640 | Convolutional 5x5x6 | 156 |
| 2 | Activation | 10 | Activation | 10 |
| 3 | Max-Pooling | 0 | Max-Pooling | 0 |
| 4 | Convolutional 3x3x128 | 73856 | Convolutional 5x5x16 | 2416 |
| 5 | Activation | 10 | Activation | 10 |
| 6 | Max-Pooling | 0 | Max-Pooling | 0 |
| 7 | Convolutional 3x3x256 | 295168 | Convolutional 5x5x120 | 48120 |
| 8 | Convolutional 3x3x256 | 590080 | Activation | 10 |
| 9 | Activation | 10 | Linear 84 | 10164 |
| 10 | Max-Pooling | 0 | Activation | 10 |
| 11 | Convolutional 3x3x512 | 1180160 | Linear 10 | 850 |
| 12 | Convolutional 3x3x512 | 2359808 | Softmax | 0 |
| 13 | Activation | 10 | | |
| 14 | Max-Pooling | 0 | | |
| 15 | Convolutional 3x3x512 | 2359808 | | |
| 16 | Convolutional 3x3x512 | 2359808 | | |
| 17 | Activation | 10 | | |
| 18 | Max-Pooling | 0 | | |
| 19 | Linear 10 | 5130 | | |
| 20 | Softmax | 0 | | |

Table 7: Architecture of Simple Convolutional Neural Networks

### A.3.2 LEARNING PARAMETERS

The parameters of the networks, both the layer weights and the coefficients of the PAUs, were trained over 100 epochs using Adam (Kingma and Ba, 2015) with a learning rate of $0.002$ or SGD (Qian, 1999) with a learning rate of $0.01$, momentum set to $0.5$, and without weight decay. In all experiments we used a batch size of 256 samples. The weights of the networks were initialized randomly and the coefficients of the PAUs were initialized with the initialization constants of Leaky ReLU, see Tab. 6. We report the mean of 5 different runs for both the accuracy on the test-set and the loss on the train-set after each training epoch.

### A.3.3 PREDICTIVE PERFORMANCE

**MNIST**

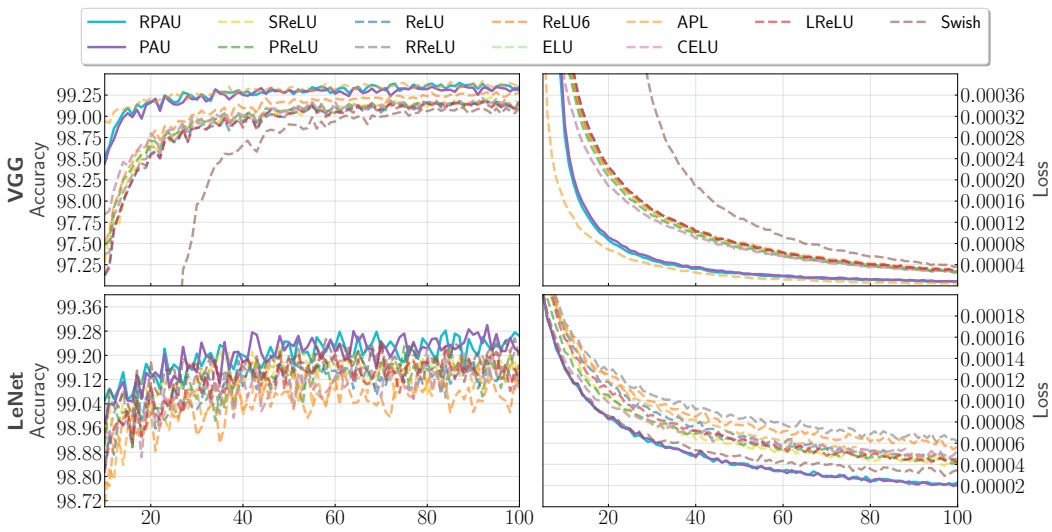

Figure 6: PAU compared to baseline activation function units on 5 runs of MNIST using the VGG and LeNet: first column mean test-accuracy, second column mean train-loss. PAU consistently outperforms or matches the best performances of the baseline activations. Moreover, PAUs enable the networks to achieve a lower loss during training compared to all baselines.

**Fashion-MNIST**

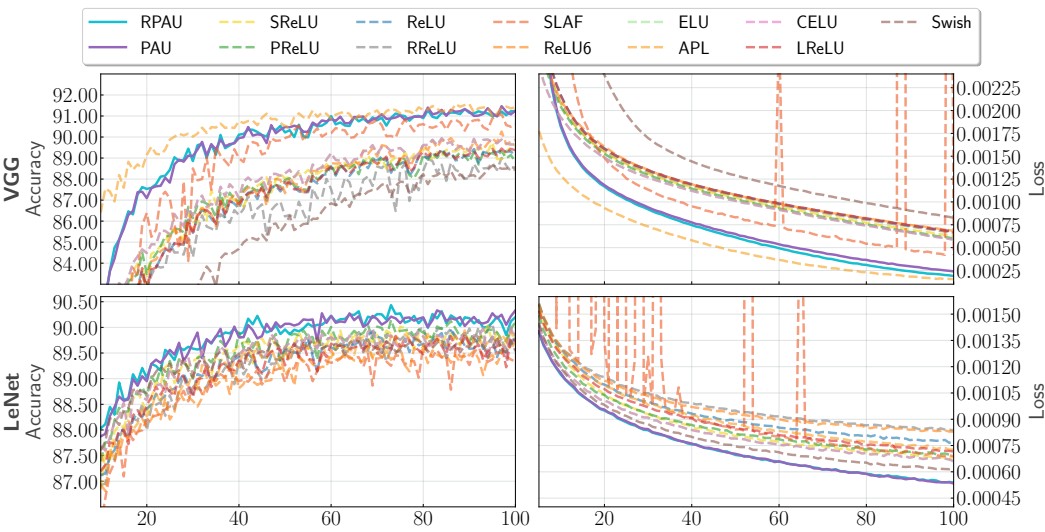

Figure 7: PAU compared to baseline activation function units on 5 runs of Fashion-MNIST using the VGG and LeNet architectures: first column mean test-accuracy, second column mean train-loss. PAU consistently outperforms the baselines activation functions in terms of performance and training time, especially on the VGG.

## A.4 DETAILS OF THE CIFAR10 AND IMAGENET EXPERIMENT

### A.4.1 LEARNING PARAMETERS

The parameters of the networks, both the layer weights and the coefficients of the PAUs, were trained over 400 epochs using SGD with momentum set to $0.9$. On the Cifar10 dataset we have different optimizer setups for PAU layers and the rest of the network. For the PAU layers we use constant learning rates per networks and no weight decay. For updating the rest of the network we use initial learning rate of $0.1$, and learning rate decay of $0.985$ per epoch and set weight decay to $5e - 4$. In all experiments we used a batch size of $64$ samples. The weights of the networks were initialized randomly and the coefficients of the PAUs were initialized with the initialization constants of Leaky ReLU, see Tab. A.1. The additive noise of the Randomized PAUs is set to $\alpha = 1\%$ training the networks VGG8 and MobileNetV2, respectively $\alpha = 10\%$ for ResNet101.

On the Imagenet dataset we use the same optimizer for PAU and the rest of the network. We follow the default setup provided by Pytorch and use an initial learning rate of $0.1$, and decay the learning rate by 10% after 30, 60 and 90 epochs.

### A.4.2 NETWORK ARCHITECTURES

The network architectures were taken from reference implementations in PyTorch and we modified them to use PAUs. All architectures are the same among the different activation functions except for Maxout. The default amount of trainable parameters of VGG8 (Simonyan and Zisserman, 2015) is 3,918,858. Using PAU 50 additional parameters are introduced. Maxout is extending the VGG8 network to a total number of 7,829,642 parameters. MobileNetV2 (Sandler et al., 2018) is contains by default 2,296,922 trainable parameters. PAU adds 360 additional parameters. The Maxout activation function is results in a total number of 3,524,506 parameters. With respect to the number of parameters ResNet101 (He et al., 2016) is the largest network we train. By default it contains 42,512,970 trainable parameters, we introduce 100 PAUs and therefore add 1000 additional parameters to the network. If one is replacing each activation function using Maxout the resulting ResNet101 network contains 75,454,090 trainable parameters. The default DenseNet121 (Huang et al., 2017) network has 6,956,298 parameters. Replacing the activation functions with PAU adds 1200 parameters to the network.

### A.4.3 PRUNING EXPERIMENT

For the pruning experiment, we implement the "Lottery ticket hypothesis" (Frankle and Carbin (2019)) in PyTorch. We compare PAUs against the best activation for the network architecture according to the average predictive accuracy from Tab. 3. More precisely, we compare the predictive performance under pruning for the networks $N_1 = \{\text{VGG-8}_{\text{pau}}, \text{MobileNetV2}_{\text{pau}}, \text{ResNet101}_{\text{pau}}\}$ against the networks $N_2 = \{\text{VGG-8}_{\text{LReLU}}, \text{MobileNetV2}_{\text{RReLU}}, \text{ResNet101}_{\text{pau}}\}$. Here we avoided Maxout as it heavily increases the parameters in the model, defeating the purpose of pruning. Unlike the original paper, we compress the convolutions using a fixed pruning parameter per iteration of $p\% = 10, 20, 30, 40, 50, 60$ and evaluated once per network. After each training iteration we remove $p\%$ of filters in every convolution and the filters we remove are the ones where the sum of its weights is lowest. After pruning, we proceed to re-initialize the network and repeat the training and pruning proceedure with the next $p\%$ parameter.

### A.4.4 Predictive Performance Cifar10

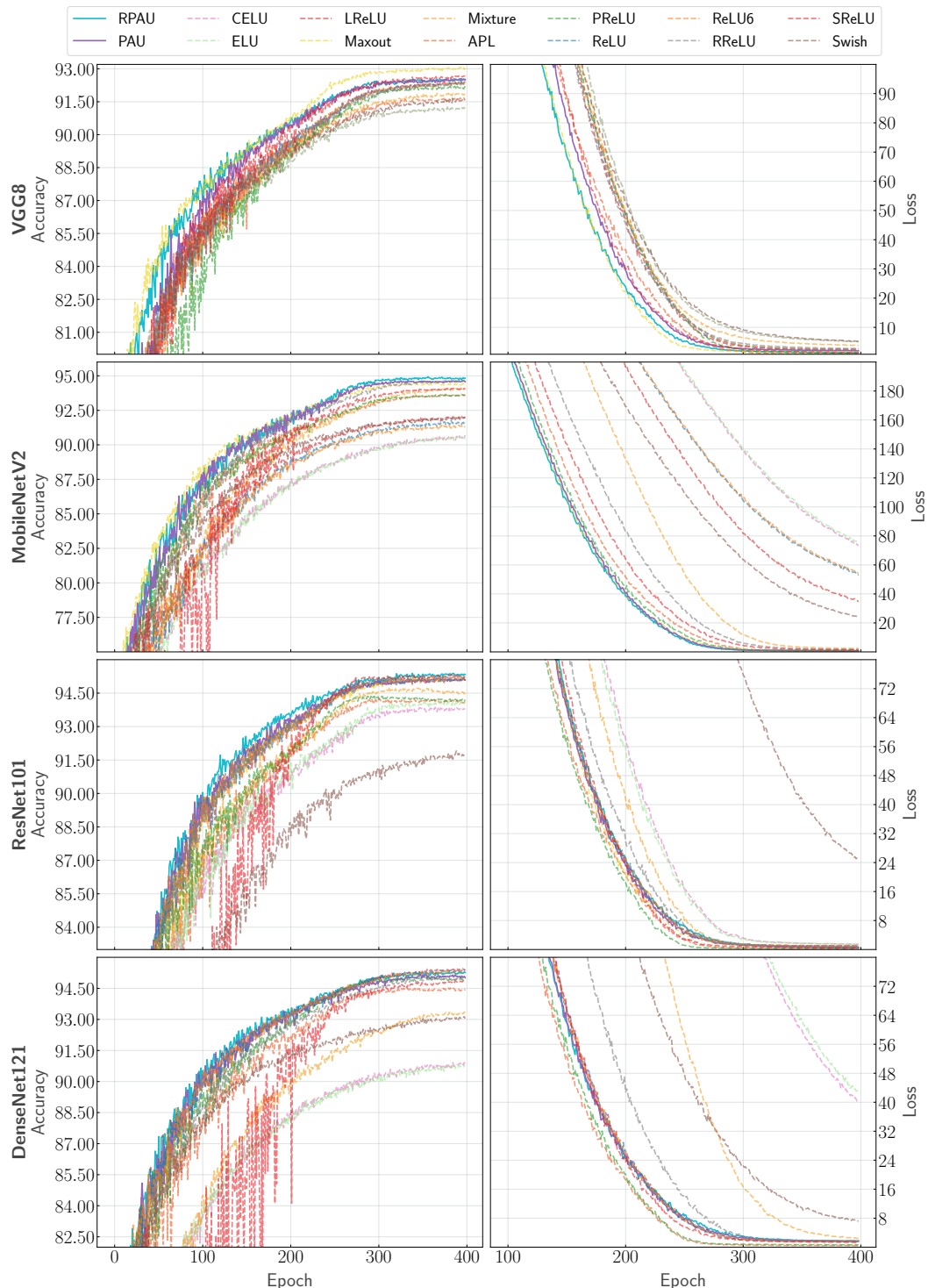

Figure 8: PAU compared to baseline activation function units on 5 runs of CIFAR-10. Accuracy on the left column and loss on the right one. (Best viewed in color)

