# OpenReview forum: "Padé Activation Units: End-to-end Learning of Flexible Activation Functions in Deep Networks"
_ICLR.cc/2020/Conference — Accept (Poster)_

### Official Review · AnonReviewer2 · 2019-10-17
**Official Blind Review #2**

**Rating:** 6

**Review:**

This work proposes an activation function that contain parameters to be learned through training. The idea is to give the learning algorithm more "freedom" to choose a good activation function, and hopefully better performance can be achieved.

The paper is well written, and the experiment results look reasonable. However, there are several key issues.

1) as the authors stated, a "good" activation function should maintain the universal approximation property of the neural network. This seems not discussed for the PADE activation function.  Does (1) satisfy the conditions (i)-(v) listed in table I? Is there a rigorous proof? Table I seems to claim that the PADE based neural network satisfies (i), but there is no formal proof.

2)  In order to avoid poles, the activation function used in this work is (2). How well can (2) approximate (1)? What is the potential loss? Perhaps there should be more discussion on this - preferably some theoretical supports.

Overall, the reviewer feels that this paper starts with an interesting idea, but the developments on the theoretical side is a bit thin.

**Experience Assessment:**

I have read many papers in this area.

**Review Assessment: Checking Correctness Of Derivations And Theory:**

N/A

**Review Assessment: Checking Correctness Of Experiments:**

I assessed the sensibility of the experiments.

**Review Assessment: Thoroughness In Paper Reading:**

I made a quick assessment of this paper.

---

> ### Author Response · Authors · 2019-11-10
> **Safe PAU Universal Approximator**
>
> We thank you for taking the time to read our paper, and the comments.
>
> It is true that we do not provide a formal proof for the safe PAU, as also mentioned by reviewer #1. However, PAU matches the assumption  of Kidger et al.  [1] proof. More precisely. Kidger et al. show that under certain size constraints, networks using non-affine continuous functions, with a continuous nonzero derivative at some point are also universal approximators. We will include this in the camera-ready version.
>
> Moreover, (i)-(v) are covered by PAU. For (i) we refer to the universal approximator discussion above as well as in the paper. Since PAU is seemingless integrated with the differentiable learning stack, standard methods for avoiding vanishing gradients can be used, hence, (ii) is covered. To cover (iii), we introduce safe PAUs and refer also to [1]. (iv) is covered since we only have a small overhead of parameters. Moreover, due to Telgarsky (2017) and  other recent results on ResNets one can actually expect to require less parameters, but this is future work. Finally (v) is covered as demonstrated by our empirical results. We will put these arguments into the camera-ready version. Thanks for pointing us to this.
>
> [1] Kidger, Patrick, and Terry Lyons. "Universal Approximation with Deep Narrow Networks." arXiv preprint arXiv:1905.08539 (2019).

---

> > ### Comment · AnonReviewer2 · 2019-11-14
> > **Thanks for the reply.**
> >
> > Thanks for the reply. I think if the universal approximation property of PAU can be rigorously shown, this paper can be much stronger. I do not mind you using Kidger et al 2019 to show it. Showing that Kidger et al covers (i)-(iv) may be straightforward, but the readers may hope to see it in a clear way.
> >
> > It is still unclear to me if the   "safe PAU" can retain the universal approximation property of PAU, since this is almost equally important. Without discussion on this point, the theoretical side still feels not significant enough.

---

> > > ### Author Response · Authors · 2019-11-14
> > > **Updates**
> > >
> > > Thanks for your comments.
> > > For (iii), we have introduced the Safe PAUs to avoid poles, which was an initial difficulty we faced when training even after few epochs on very simple networks. To prove some guarantees regarding exploding values, we could introduce a form of Lipschitz regularization that combined with BatchNorm could give us some initial assumptions.
> > > Regarding a proof against vanishing gradients (ii), this could be connected with a relaxed version of the Safe PAUs where the denominator is allowed to be < 1, as mentioned in the paper, this could potentially allow for gradient amplification.
> > > For the moment, we show empirically that Safe PAU is stable and doesn’t suffer from vanishing or exploding gradients more than any of the other activation functions that we compared to.
> > >
> > > More importantly, we have updated the paper and included the proof for (i), we also added the calculation for the number of parameters (iv) as $\phi=L*(m+n)$, which for our experiments is $\phi=10*L$ where $L$ is the number of activation layers. The exact number of parameters for the experiments are in the Appendix, and they are orders of magnitude less than the remaining parameters of the network.
> > >
> > > We thank you for the discussion and motivation to make the paper stronger.

---

### Official Review · AnonReviewer1 · 2019-10-22
**Official Blind Review #1**

**Rating:** 8

**Review:**


This paper introduces a novel parametric activation function, called the Pade Activation Unit (PAU), for use in general deep neural networks. Pade is a rational function, which is a ratio of two polynomials, and which can very well approximate any of the usually used activation functions while having only a few parameters that can be learned from data. Moreover, the authors identify five properties that an activation function should have, and either prove or empirically show that PAUs satisfy all of them, unlike some of the baselines. Additionally, since Pade approximation can have poles and be unstable, this work introduces safe PAUs, where the polynomial in the denominator is constrained to attain values greater than or equal to one. Since one of the suggested properties is that a function using a given activation function be a universal function approximator, the authors provide a sketch of a proof that PAUs do allow that. This proof applies only to the unsafe version of the PAU, and it is unclear whether it extends to the safe PAU---an issue that is not mentioned by the authors.
Furthermore, the authors propose a stochastic version of PAU with noise injected into parameters, which allows regularization. The empirical evaluation is quite extensive, and the PAU is compared against nine baselines on five different architectures (LeNet, VGG, DenseNet, ResNet, MobileNet) on four different datasets (MNIST, Fashion MNIST, CIfar10, ImageNet) for the classification task. The evaluation confirms that PAUs can match the performance of or sometimes outperform even the best baselines while the attained learning curves show that PAUs also lead to faster convergence of trained models. Finally, the authors demonstrate that (and provide intuition why) using PAUs allow for high-performing pruned models.

I recommend ACCEPTing this paper as it is well written, extensively evaluated, and provides performance improvements or at least matches the performance of the best baseline across several datasets and model architectures.

My only two suggestions for improvement are a) make the universal approximation proof tighter by making sure that it extends to the safe PAU version, and b) evaluate the proposed activation function on tasks other than just classification.

**Experience Assessment:**

I have read many papers in this area.

**Review Assessment: Checking Correctness Of Derivations And Theory:**

I carefully checked the derivations and theory.

**Review Assessment: Checking Correctness Of Experiments:**

I carefully checked the experiments.

**Review Assessment: Thoroughness In Paper Reading:**

I read the paper at least twice and used my best judgement in assessing the paper.

---

> ### Author Response · Authors · 2019-11-10
> **Safe PAUs**
>
> We thank you for the time and comments.
>
> Indeed, the evaluation of the experiments is quite expensive and this is one of the challenges we are facing. Although we intend to test PAUs on other tasks/architectures, we consider the comparisons to the baselines we have (including the experiments proposed by reviewer #3) as a good introduction for PAUs into the community.
>
> You are right in that we do not have a proof for the safe version of PAUs presented in the paper, but we are equally interested in this topic, too. Consequently, we now tested another “safe” version of the form P(X)/(eps + |Q(X)|). This version can be proven to be a universal approximator via similar arguments as the general PAU. However, empirically this version turned out to be very unstable. Unfortunately, the existence and form of safe PAUs, does not necessarily tell us about the stability and optimization characteristics. Fortunately, there is an indirect way around this and we redirect you to our reply to reviewer #2 for a further discussion.
>
> Thank you once again for your review.

---

> > ### Comment · AnonReviewer1 · 2019-11-13
> > **I think it's a good paper.**
> >
> > Thanks for the response. I am not convinced by your argument about safe PAUs above, but I like the additional experiments performed in response to R3. I'm keeping my scores and am recommending acceptance.

---

### Official Review · AnonReviewer3 · 2019-10-23
**Official Blind Review #3**

**Rating:** 6

**Review:**

The authors introduce an activation function based on learnable Padé approximations. The numerator and denominator of the learnable activation function are polynomials of m and n, respectively. The authors name them Padé activation units (PAUs). The authors also propose a randomized a version of these functions that add noise to the coefficients of the polynomials in order to regularize the network. The authors show, at best, marginal improvements over a variety of baselines including MNIST, fashion MNIST, CIFAR10, and Imagenet. The authors also show that pruning neurons with PAU units results in slightly better accuracy that pruning neurons with ReLU units.

The improvements over baselines shown were marginal and I do not think they warrant publication at this conference. The accuracy improvements were no more impressive than other learned activation functions which the authors perhaps did not see, such as SReLUs (Deep Learning with S-Shaped Rectified Linear Activation Units) and APLs (Learning Activation Functions to Improve Deep Neural Networks).

** After author response **
Changing from reject to weak accept
The authors have included new experiments that compare to a wider range of learned activation functions. While not ground breaking, it shows that it is competitive with state-of-the-art learned activation functions and could have something to offer.

**Experience Assessment:**

I have published one or two papers in this area.

**Review Assessment: Checking Correctness Of Derivations And Theory:**

I assessed the sensibility of the derivations and theory.

**Review Assessment: Checking Correctness Of Experiments:**

I carefully checked the experiments.

**Review Assessment: Thoroughness In Paper Reading:**

I read the paper at least twice and used my best judgement in assessing the paper.

---

> ### Author Response · Authors · 2019-11-10
> **Further Experiments**
>
> We thank you for the time and comments.
>
> We have a different, rather very positive perspective. We are proposing an activation function that helps practitioners to avoid the search for activation functions as done in [1], and replaces this by learning.  PAU can match the performance of or sometimes outperform even the best baselines, in some cases up to 2% better than common activation functions. For instance we are  boosting the performance of MobileNetV2, a CVPR 2018 state-of-the-art approach. Moreover, in contrast to previous work, PAU directly paves the way to provably robust deep learning (Croce et al., 2019). Nevertheless, we fully agree with the reviewer that the paper would be even stronger by comparing to more learnable activation functions such as SReLUs and APLs, so we implemented and ran some more experiments:
>
>
>                   MNIST_VGG        MNIST_LeNet
> SReLUs    $99.15 \pm0.03$    $99.13 \pm0.14$
> APLs        $99.18 \pm0.10$    $99.35 \pm0.11$
> PAU        $99.30 \pm0.05$    $99.21 \pm0.04$
>
>                   FMNIST_VGG        FMNIST_LeNet
> SReLUs    $89.65 \pm0.42$    $89.83 \pm0.30$
> APLs        $91.41 \pm0.48$    $89.72 \pm0.30$
> PAU        $91.25 \pm0.18$    $90.30 \pm0.15$
>
>
>                   CIFAR10_VGG        CIFAR10_MVNet      CIFAR10_RNet
> SReLUs    $92.66 \pm0.27$    $94.03 \pm0.11$    $95.24 \pm0.13$
> APLs        $91.63 \pm0.13$    $93.62 \pm0.64$     $94.12 \pm0.36$
> RPAU        $92.50 \pm0.09$    $94.82 \pm0.21$     $95.34 \pm0.13$
>
>
> Again, of the 7 new experiments, PAU is better than APL in 5 of them, and better than SReLU in 6 of them. More experiments on DenseNet and ImageNet, are running, and we expect to have them before the rebuttal deadline is over. Hence, PAU’s perspective on robust deep learning via rationalization gets even more interesting. Thanks for pushing us to run more experiments. We believe the experiments show that PAUs are indeed competitive and have a place among the learnable activation functions.
>
> We will keep you posted.
>
> [1] P. Ramachandran, B. Zoph, and Q. V. Le. Searching for activation functions. In Proceedings of the Workshop Track of the 6th International Conference on Learning Representations (ICLR), 2018.

---

> > ### Author Response · Authors · 2019-11-14
> > **Further Experiments**
> >
> > We have a small update: Here are the remaining experiments which will finish before the Deadline for Author Comments and Responses.
> >
> >               	     CIFAR10_Densenet
> > APL*                 $94.45\pm0.23$
> > SReLU		  $94.77\pm0.24$
> > RPAU                $95.27\pm0.10$
> >
> >
> >               Imagenet_MobileNetV2
> > 			Acc@1	  Acc@5
> > SReLU	      $70.62$   	 $89.59$
> > Swish            $71.24$     $89.95$
> > PAU	      $71.35$	 $89.85$
> >
> > SReLU outperforms most of the activation functions. However, both Swish and PAU outperform SReLU.
> >
> > The following table shows the summary of our experiments:
> >
> > 					    | ReLU | ReLU6 | LReLU | RReLU | ELU | CELU | PReLU | Swish | Maxout |Mixture| APL | SReLU |
> >
> > PAU/RPAU >= Baseline   | 33       | 34        | 33        | 32 	| 39    | 39 	 | 38 	|  41       | 9 	         | 20 	 | 32    | 33	   |
> > PAU/RPAU < Baseline     |   8       |   7        |   8        |   9         |   2    |  2        |  3 	|    1       | 6 	         |  0 	 |   7    |   8 	   |
> >
> > Again, we believe the experiments show that PAUs are indeed competitive and have a place among the learnable activation functions.

---

### Decision · Program_Chairs · 2019-12-19

**Decision:**

Accept (Poster)

**Comment:**

The paper proposed a new learnable activation function called Padé Activation Unit (PAU) based on parameterization of rational function. All the reviewers agree that the method is soundly motivated, the empirical results are strong to suggest that this would be a good addition to the literature.